# Digital Biomarkers for Personalized Nutrition: Predicting Meal Moments and Interstitial Glucose with Non-Invasive, Wearable Technologies

**DOI:** 10.3390/nu14214465

**Published:** 2022-10-24

**Authors:** Willem J. van den Brink, Tim J. van den Broek, Salvator Palmisano, Suzan Wopereis, Iris M. de Hoogh

**Affiliations:** Netherlands Organisation for Applied Scientific Research (TNO), 2333 BE Leiden, The Netherlands

**Keywords:** digital biomarkers, personalized nutrition, continuous glucose monitor (CGM), wearables, meal detection

## Abstract

Digital health technologies may support the management and prevention of disease through personalized lifestyle interventions. Wearables and smartphones are increasingly used to continuously monitor health and disease in everyday life, targeting health maintenance. Here, we aim to demonstrate the potential of wearables and smartphones to (1) detect eating moments and (2) predict and explain individual glucose levels in healthy individuals, ultimately supporting health self-management. Twenty-four individuals collected continuous data from interstitial glucose monitoring, food logging, activity, and sleep tracking over 14 days. We demonstrated the use of continuous glucose monitoring and activity tracking in detecting eating moments with a prediction model showing an accuracy of 92.3% (87.2–96%) and 76.8% (74.3–81.2%) in the training and test datasets, respectively. Additionally, we showed the prediction of glucose peaks from food logging, activity tracking, and sleep monitoring with an overall mean absolute error of 0.32 (+/−0.04) mmol/L for the training data and 0.62 (+/−0.15) mmol/L for the test data. With Shapley additive explanations, the personal lifestyle elements important for predicting individual glucose peaks were identified, providing a basis for personalized lifestyle advice. Pending further validation of these digital biomarkers, they show promise in supporting the prevention and management of type 2 diabetes through personalized lifestyle recommendations.

## 1. Introduction

Type 2 diabetes (T2D) is a top-10 leading cause of disability-adjusted life years (DALYs) in the last decade, and it is anticipated to affect more than 7% of the world population by 2030 [1,2]. Beyond pharmacological therapy, lifestyle medicine targeting insulin resistance as the root cause of T2D is becoming evident now in remitting, reversing, or preventing the disease [3,4,5,6,7,8]. Digital technologies that support individuals in changing and monitoring their lifestyles, such as dietary behavior, physical activity, sleep, and stress, are promising for supporting lifestyle medicine [3].

The implementation of lifestyle medicine with sustained lifestyle behavior change necessitates a personalized approach, including personalized diagnosis and diet, physical activity and stress management, self-empowerment, motivation, participation, and health literacy [3]. Increasing evidence shows that T2D subgroups exist with different underlying etiology, demonstrating a differential response to lifestyle interventions [9,10,11,12,13]. Additionally, several studies have demonstrated the potential of a personalized nutrition approach to improve health in a (relatively) healthy population [14,15,16,17,18]. Full remission into a healthy glucose metabolism through lifestyle medicine is well achievable, especially in the early phase preceding the disease. Multiple studies, indeed, have shown that lifestyle medicine is only successful in achieving T2D remission in a pre- or less advanced stage of the disease, but often fails in persons who have a more advanced, irreversible stage of T2D, especially those with β-cell dysfunction or combined tissue insulin resistance [19,20,21]. Therefore, early diagnosis and intervention are essential for reducing the societal burden of T2D. In most of these studies, an extensive baseline assessment, including invasive measurements, such as blood, saliva, or feces collection and postprandial biomarker evaluation with challenge testing, was used to provide personalized dietary recommendations. Challenge tests, such as a mixed-meal challenge test or an oral glucose tolerance test (OGTT), offer insights into dynamical biomarker responses to a standardized meal, as opposed to solely looking at overnight fasting biomarkers [22]. This allows for earlier detection of a pre-stage of the disease or derailment of health. T2D develops gradually, whereas prediabetes can exist for years with increased levels of insulin but relatively normal levels of overnight fasting glucose [23].

Wearable technologies, including smartphones and smartwatches, are increasingly utilized in the healthcare domain for the development of so-called digital biomarkers [24,25,26]. This novel type of biomarker is characterized by being measured non-invasively, continuously, and under real-world conditions using digital technology, allowing for a more holistic and personal insight into someone’s health. Therefore, digital biomarkers enable accessible health and behavioral feedback to the user and are particularly suited for driving the healthcare transition towards prevention, empowering people in the self-management of health and disease [27]. Additionally, continuous, non-invasive, or minimally invasive measurements may allow for the measurement of subtle health derailments by evaluating the responses or resilience towards daily challenges or perturbations, thereby allowing for such early diagnosis [27]. Continuous glucose monitoring (CGM), for example, is used to define so-called ‘glucotypes’ based on glucose patterns, which are associated with clinical biomarkers of glucose dysregulation [28]. Furthermore, digital biomarkers can provide users with more frequent and detailed contextual information and continuously update personal lifestyle recommendations. Indeed, postprandial glucose responses to meals are highly personal and depend on a person’s genetic makeup and clinical factors (e.g., BMI, microbiome, lipid levels), and also on the context of the meal, and include factors such as sleep, physical activity, and composition of previous meals [29,30]. These contextual factors, which strongly influence glucose homeostasis, are difficult to manage in a traditional healthcare setting due to their limited ability to capture daily life conditions. Recently, it was shown that interstitial glucose levels can be predicted from continuous contextual data, including those on diet and physical activity, in persons with prediabetes under real-world conditions [31]. High-quality contextual data collection is essential for this. While, for sleep and activity tracking, wearable technologies are becoming more reliable for passive monitoring, meal tracking requires active, continuous logging from the user, impacting this essential data quality. Food frequency questionnaires, 24 h recall interviews, or food diaries are the most common methods for monitoring dietary behavior and estimate dietary intake, although these methods are susceptible to misreporting [32,33]. Recently, CGM-based meal-detection algorithms were proposed for people with type 1 diabetes (T1D), showing the potential for CGM to support dietary intake monitoring [34,35]. To our knowledge, this has not been presented for healthy people, persons with prediabetes, or persons with T2D. Altogether, there is a need for high-quality contextual data from everyday life that can be linked to glucose dynamics to support health self-management for the prevention of T2D.

The current study set out to demonstrate a proof of principle for detecting eating moments with CGM, as well as predicting and explaining glucose levels based on contextual factors, such as sleep, activity, and diet in a personalized manner, ultimately supporting health self-management and prevention of T2D. Therefore, we performed an observational study with 24 healthy adult volunteers who conducted continuous self-monitoring for two weeks in a real-life setting. The volunteers wore a CGM device for glucose monitoring and a smartwatch for monitoring physical activity and sleep and logged their food intake via a mobile food diary app. The study evaluated how well the probability of having an eating moment can be predicted based on continuously measured glucose, sleep, and activity data. Personalized insight into eating moments can form the basis for personalized advice on the timing of eating. In addition, passive detection of eating moments has the potential to notify individuals to fill out the food diary and to improve compliance with data collection. Additionally, personal glucose prediction models were created to model the effects of physical activity, dietary intake, and sleep on individual glucose levels. How well the glucose levels can be predicted based on personal lifestyle behavior, including dietary intake, sleep, and activity data, was evaluated since personalized insights into the effects of lifestyle behavior on glucose levels can support an individual in keeping glucose levels within a healthy range.

## 2. Materials and Methods

### 2.1. Study Design and Data Collection

Twenty-four healthy volunteers with an affinity for nutrition and health research were included in this observational pilot study. Participants were eligible for study participation if they were aged 20–65, owned a smartphone, and had a finger-prick glucose value < 7.8 mmol/L after eight hours of fasting during screening to exclude for unknown type 2 diabetes. Exclusion criteria were having type 2 diabetes, body mass index > 30 kg/m^2^, and conditions that would not allow the use of a continuous glucose monitoring system, such as a skin allergy or eczema. All participants gave written, informed consent.

At inclusion, participants were equipped with self-monitoring devices, installed the custom-built Android- and iOS-compatible *HowAmI* app (TNO, Zeist, The Netherlands) on their smartphones, and were instructed in the use of all devices and apps. The study consisted of 14 days of self-monitoring in a real-life setting. The self-monitoring devices were the Abbott FreeStyle^®^ Libre™ Pro (Abbott GmbH & Co, Wiesbaden, Germany) continuous glucose monitoring (CGM) device and the Philips Elan wristband (Koninklijke Philips N.V., Eindhoven, The Netherlands). The factory-calibrated CGM device was worn on the upper arm and measured subcutaneous interstitial glucose concentrations every fifteen minutes. Participants were blinded to their glucose values. Glucose data were stored on the CGM devices, which were collected at the end of the study. Glucose measurements from the first day were excluded to allow for the stabilization of the sensor. The Elan wristband collected data via a raw green spectrum photoplethysmogram (PPG) sensor and accelerometer. Data were regularly offloaded by participants using ElanControl software (Koninklijke Philips N.V., Eindhoven, Netherlands) and transferred to Philips after the study. Proprietary algorithms were used to translate the raw data into sleep, energy expenditure, ACN, and heart rate. The *HowAmI* app was used for collecting food intake. The app was custom-built to provide the functionality to record the exact date and time of the recorded meals. The *HowAmI* app uses the *MyFatSecret* food database and back-end (Secret Industries Pty Ltd., Victoria, Australia) to record food intake and connects to a custom, parallel back-end database to record the time stamp for each meal. This same database was used to collect and store data from the continuous glucose monitor. Helpdesk support was available throughout the study. Participants could follow their regular lifestyle during the study. The study protocol was approved by the Medical Ethics Committee Brabant (NL68969.028.19). The study was performed in accordance with the Declaration of Helsinki and good clinical practice and registered at the Netherlands Trial Register: NL7117.

### 2.2. Data Preprocessing and Feature Engineering

After data collection, all subsequent data processing, analysis, and visualization were performed using R, version 4.1.2. We used packages ggplot 3.3.5, xgboost 1.5.0.2, caret 6.0–90, pracma 2.3.8, and treeshap 0.1.1 [36,37,38,39,40]. Figure 1 provides a schematic overview of the different steps taken in data preprocessing, model fitting, and model analysis, while details are provided below.

We created an appropriate dataset for the detection of meals in a multi-step process consisting of data aggregation and feature engineering. Several engineered features were created from continuous glucose sensor data that matched the sampling interval of the glucose sensor. We created lag, lead, the difference of the lead (1st to 6th order), the difference of the log lead (1st to 6th order), lagged difference of the lead (1st to 6th order), standard deviation and mean of the lead up to 90 min, standard deviation and mean of the lag up to the 90 min, standard deviation and mean from a 90 min lag to a 90 min lead, relative standard deviations of the 90 min lead and lag, the ratio between the standard deviation of the lead and the lag, and the ratio between the 90 min lead and lag maximum and minimum values. No other modalities were used in the meal detection dataset. The target variable was given as a classification label, where the positive class denotes that food was taken at that respective time point, and the negative class denotes that no food was taken. A time point was considered to be of the positive class if the meal contained any carbohydrates; the time points immediately preceding and following food intake were also considered to be of the positive class to account for inaccuracies in diary annotation and the time it took to consume the food.

A similar approach was taken to create a suitable dataset for the prediction of glucose. Data were first aggregated to deal with varying sampling intervals across the different modalities. The items of any meal that were eaten within 15 min were combined to form a single meal. For each meal, total calories, as well as calories from fat, protein, and carbohydrates, were calculated. From this, the fractions of calories from fat, protein, and carbohydrates were also derived. Additionally, energy expenditure, acceleration (movement), and heart rate features collected from the Elan wristband were aligned to the collection interval of the continuous glucose sensor (once every 15 min) and then aggregated to match the frequency of the glucose measurements before being joined. Sleep and sleep stage information were subsequently joined so that each glucose value was associated with sleep feature values from the closest preceding period of sleep, but no more than 28 h earlier.

Subsequently, we created new features from all aggregated and joined data (except sleep-related data) by averaging the values for all features over rolling periods of 30, 60, and 90 min as well as 2, 3, 8, and 24 h. In the case of caloric intake, energy expenditure, and activity, these features were created by taking the rolling sum instead of the rolling average.

The final datasets were created from this data by selecting the longest stretch of uninterrupted data that was available from every included participant with a minimal stretch of 24 h. For this purpose, we defined ‘uninterrupted data’ as periods where data from all modalities were available without a break in the glucose sensor measurements and no break in the activity or food intake measurements. In addition, the total length of combined stretches per participant needed to exceed 7 days to allow sufficient data for training and test sets.

The training set for the glucose prediction model contained all available data for each participant apart from those from the last 3 days; these were kept separate for the testing set. The training dataset for the meal detection model was more limited; we used the first 4 days of data for each participant for training purposes and kept a subsequent 3-day period as the training set. The training dataset for the meal detection model was kept intentionally smaller to imitate a practical situation where only limited data can be obtained because of the participant burden of keeping a food intake diary.

In the case of both models, the test dataset was used to estimate model generalizability but not for any other purposes.

### 2.3. XGboost for Predicting Eating Moments and Glucose

For both models, we used recursive feature elimination (RFE) with 10-fold cross-validation to obtain the smallest set of features that would still perform similarly to the full feature set. This step was undertaken to simplify the model for easier interpretation and reduce overfitting. In this procedure, we used xgboost as the underlying model to drive feature selection; *gain*, as a measure of improvement in accuracy, was used to rank feature importance during RFE [37]. A fixed number of rounds (100) was used at every iteration of the procedure; no hyperparameter tuning was performed. The smallest set of features where model performance was within 10% of the best-performing set was selected for use in the final prediction model.

For the final glucose prediction model, the model hyperparameters were tuned by minimizing the mean absolute error using random search with 10-fold cross-validation. The target variable was the log-transformed glucose value. The hyperparameter tuning procedure for the meal detection model minimized the classification error using grid search with 10-fold cross-validation. Grid search was chosen over random search because of the propensity for overfitting and its reduced computation time because of a smaller amount of training data compared to the glucose prediction model.

For the glucose prediction model, Shapley values were calculated for all data points in the training dataset by using the implementation of the algorithm described in Lundberg et al. [41] and provided by the *treeshap* package. This algorithm was defined using a mathematical game theoretic approach that is explained in detail by Lundberg et al. [41]. These Shapley values provided information for each predicted value about the influence of each model feature in making that prediction. We used the Shapley values to determine overall feature importance (Appendix A) by taking the mean absolute Shapley value for each feature for all predictions. Furthermore, we used the Shapley values to determine the feature influence in the prediction of peak glucose levels. Using the *findpeaks* algorithm of the *pracma* package, we identified peaks for all subjects where the glucose value was higher than at least the $IQR+Q_{3}$ for that subject. This led to a varying number of identified peaks for each of the subjects; for further analysis, we included only those subjects with 10 or more identified peaks.

## 3. Results

### 3.1. Baseline Characteristics and Dataset Characteristics

A total of 24 individuals participated in the study. The study participants were, on average, 39 +/−12 years old with an average body mass index (BMI) of 22 +/−9.4 kg/m^2^. Of the 24 participants, 17 participants were female (71%). All individuals had non-fasting blood glucose in the normal range below 7.8 mmol/L during screening, which excluded people with unknown diabetes. A minimal length of uninterrupted data periods for 24 h from all continuous data sources was selected for each individual to guarantee sufficient data quality. Additionally, at least three days of test data and three days of training data were required to ensure sufficient power to perform the analysis. Applying these two criteria resulted in a dataset with 11 individuals with 4–11 days of training data and three days of testing data that was selected for further analysis (Figure 2).

### 3.2. Detecting Eating Moments Based on Interstitial Glucose Levels

Detecting eating moments can support food logging, for example, through AI-driven notifications, thereby reducing the risk of erroneous reporting. We developed an extreme gradient boosting machine model to predict the probability of having an eating moment in healthy individuals. Eating moments were predicted in segments of 30 min based on continuously collected interstitial glucose, sleep, and activity data over three days per participant. After model training, accuracy, specificity, and sensitivity were calculated using a hold-out test dataset. The final model showed an accuracy, specificity, and sensitivity of 92.3% (87.2–96%), 98.9% (97–100%), and 90.8% (86.4–94.9%), respectively. The accuracy, specificity, and sensitivity in the test dataset of another three days per individual were 76.8% (74.3–81.2%), 60.3% (33.3–82.6%), and 78.4% (74.3–84.1%), respectively. Figure 3 visualizes the predicted probability in segments of 30 min against the observed eating moment for both the training and the test dataset, confirming the high level of accuracy in the training dataset. The test dataset, however, presented lower accuracy.

### 3.3. Predicting Lifestyle Behavior Effects Based on Interstitial Glucose Levels

The glycemic response is highly personal, depending on biological and contextual factors, such as lipid metabolism, muscle mass, nutrition, stress, activity, and sleep. The individual glycemic response may, thus, vary between and within individuals. Here, we applied an extreme gradient boosting machine approach with the subject number as a random variable to allow personalized models to predict glucose levels from contextual factors in real time. Continuous glucose levels for three days were predicted from 72 features engineered around nutrition, activity, and sleep over different periods (short term: 3 h, long term: 8 h, and 24 h). An overall mean absolute error (MAE) of 0.32 (+/−0.04) mmol/L for the training data and 0.62 (+/−0.15) mmol/L for the test data was obtained. Figure 4 shows an example of the goodness of fit for subject 09 from the training dataset (Figure 4A) and the test dataset (Figure 4B). Bland–Altman analysis indicated a bias lower than 0.01 mmol/L in both the training and the test set and 2.5 and 97.5 percentile limits of agreement ranging from −0.72 to 1.1 in the training set and from −1.56 to 1.8 mmol/L in the test set (Figure 5).

The final model contained 17 features after feature selection, covering activity, nutrition, sleep, and unexplained, subject-specific features. The influence of the different features is summarized in Table 1, and the details are specified in Appendix A. The influence of cardiometabolic factors was 26.7%, while the contribution of the unexplained, subject-specific features was 24.1%. The influence of short- and long-term activity was 10.9% and 12.5%, respectively. The short- and long-term nutrition features had an influence of 10.7% and 8.7%. Finally, the contribution of the sleep features was 10.7%. Although these numbers indicate an overall insight into the importance of the features in predicting glucose levels, they may have been very different between and within individuals across the study period. To provide personalized insights into the relationship between the contextual factors and glucose levels, we applied the SHAP (Shapley additive explanations) procedure to the selected model. With the goal of personalized insight being to reduce high glucose peaks, high glucose peaks were identified for each participant. Shapley values were then calculated for each of those glucose peaks to determine the feature influence for those specific glucose data points. Figure 6 shows the frequency of the five most important features per data point per participant when explaining their highest peaks. Overall, there was no specific category of features that was important for explaining the highest peaks, but, at the individual level, some features occurred more frequently. For example, sleep duration was never important when explaining the glucose peaks of subjects 15 and 19, while, for subjects 9 and 10, it was a relatively frequent feature. As another example, the glucose peaks of subject 22 were most often explained by their energy expenditure over the last 24 h and their subject-specific model intercept. This may indicate that the glucose peaks for this participant were related to the subject-specific variance that remains unexplained by the features thus far included in this model.

## 4. Discussion

In this study, we aimed to prove the feasibility of using real-life CGM data combined with contextual data to make predictions on an individual basis in healthy persons. First, we showed the ability to predict eating moments using interstitial glucose. Second, we showed the ability to predict and explain current interstitial glucose values using contextual data, including those relating to food intake, physical activity, and sleep.

### 4.1. Meal Detection

Dietary intake assessment is challenging since common methods are susceptible to misreporting. Several technological innovations currently focus on image recognition of meal photographs, eating action detection, and biochemical sensors that are targeted at specific nutrition-associated metabolite concentrations in non-invasive biofluids such as urine and sweat [42]. These innovations cover different aspects of dietary intake assessment, ranging from quantifying meal composition and intake of specific nutrients to meal timing. Here, we used continuous glucose monitoring with a gradient boosting machine algorithm to predict eating actions. The resulting model showed excellent performance on the training set, with an accuracy, specificity, and sensitivity of more than 90%. For the test set, the performance was moderate to good, with an accuracy of 76%, specificity of 60%, and sensitivity of 78%. The reason that the test dataset presented with a lower accuracy was possibly related to variable behavior or inconsistent food logging within the individuals—indicated by highly variable kcals/day recorded by individuals—making the model not fully generalizable across the full study period. Future work could include a reinforcement learning approach to continuously update the algorithm specifically for an individual. Otherwise, a more controlled approach against a ground truth reference, for example, with video camera monitoring of eating moments, may be applied to further investigate this and improve the algorithm upfront. To our knowledge, this is the first study demonstrating a prediction algorithm for eating moments in a healthy population in a real-life setting. Most studies on meal detection using CGM data so far have focused on T1D, with the potential for automated timing of insulin administration, for instance, in an artificial pancreas [34,35,43]. Sensitivity rates in some of these studies, if reported, were higher compared to the sensitivity of our model, but, as the glucose response to meals in people with T1D is faster and higher as there is no compensatory action from insulin, these results cannot directly be compared. The potential application of meal detection in a healthy or a (pre-)T2D population is different and may, therefore, require different levels of accuracy, sensitivity, and specificity than those required in the case of medical purposes. In T1D patients, meal detection is applied to control insulin administration, whereas, in a healthy or a (pre-)T2D population, meal detection can be used to provide individuals with more insight into their eating behavior and may provide opportunities for personalized feedback on frequency or timing of eating moments. In the future, it may even be possible to predict both meal timing and dietary composition from CGM data [44], which would provide even more opportunities for personalized advice to stimulate behavior change. Meal detection algorithms could also play a role in improving the quality of food diary applications. The collection of food intake data is known to be subject to misreporting [33]. Active recall using notifications via a smartphone app after the detection of a meal moment, could, for example, aid in improving compliance with food intake data collection.

### 4.2. Predicting Glucose

Personalized nutrition is gaining momentum in science to support health maintenance and disease prevention, especially prevention of chronic, lifestyle-related diseases such as type 2 diabetes [29,30,45]. While personalized nutrition approaches still require relatively invasive measurements in a standardized clinical setting, here, we set out an approach that allows personalized nutrition monitoring in everyday life using CGM, activity tracking, sleep monitoring, and a food diary. For predicting glucose levels using contextual data, we engineered 72 features from physical activity, meal composition, and sleep data, which were used to train an extreme gradient boosting algorithm. We engineered both short- and long-term features for physical activity and nutrition, as research has shown that both physical activity and nutrition have an acute as well as a more long-term effect on glucose levels [46,47,48,49,50]. The recursive feature elimination (RFE) step provided a subset of features by eliminating features with redundant information. This subset provided similar final model performance as when all features were included. This reduction in the number of features aided the interpretation of the final model and decreased training times. The feature selection method influenced which feature became part of the final model, and, as such, the final model did not cover all possible relationships of the full set of features with the glucose response. The choice of the feature selection algorithm, therefore, was an important consideration regarding the final result.

The overview of the final, overall feature influence confirmed the importance of physical activity, dietary intake, and sleep in determining glucose values [51,52,53,54]. In the overall model, physical activity and nutrition had a comparable influence on interstitial glucose values (Table 1). Research in T2D has, indeed, shown that structural, physical activity of more than 150 min per week is associated with a greater decline in HbA1c than lower amounts of physical activity [55]. Alternatively, the long-term physical activity features in our model may also serve as a proxy for prolonged sedentary behavior, which has also been associated with higher glucose values [56,57]. Future development of activity tracking should explicitly separate physical activity from sedentary behavior to improve personalized insight into their relation to glucose concentrations. Interestingly, two other studies showed a larger contribution of meal composition than that of physical activity, while comparable features for nutrition were used (number of calories, protein, sugar, fat, and carbohydrates over a specific time window) [30,58]. While the importance of physical activity in our model and that of Bent et al. was comparable (17% and 19%, respectively), the influence of nutrition was lower in our model (20% and 37%, respectively) [58]. Possibly, this is explained by the fact that they focused on persons with prediabetes only, while the current study targeted a healthy population. Although it should be noted that prediabetes was not excluded, 16 out of 24 participants had a fasting plasma glucose below 5.6 mmol/L. As persons with prediabetes are already insulin resistant, a higher postprandial glucose response after consumption of carbohydrate-rich foods compared to that of a healthy population is to be expected. Berry et al. also indicated greater influence of nutrition, while including healthy people [30]. However, in their study, only the effects of standardized meals over a short time frame were investigated, and subjects were instructed to limit exercise on test days. This may explain why meal composition as compared to physical activity was more important in their model. Finally, sleep, as a lifestyle-related factor, had a significant influence on interstitial glucose concentrations, albeit less than nutrition and physical activity (11%). This is in concordance with the aforementioned personalized nutrition studies investigating the influence of contextual factors on glucose control [30,31]. Indeed, sleep disturbance is linked to impaired glucose control, while sleep interventions may contribute to its normalization [59]. In addition to contextual lifestyle factors, cardiometabolic features (energy expenditure, average heart rate) and an unexplained, subject-specific feature were identified as influencing glucose levels. This confirms previous findings showing that there is a large interindividual variability in glucose response, which can only partly be explained by measured contextual factors [29,31,60]. Adding other factors such as psychological stress, genetics, metabolic health, cardiovascular health, anthropometry, and demography may further increase the predictive power of the model [61,62,63]. However, as the relative contribution of the unexplained, between-person variation was less than 25%, one may want to be mindful of adding burdensome or expensive measurements such as genetics and blood biomarkers considering their probable limited impact on the model.

The strength of this study is the real-world design, maximizing the ecological validity of the observations. However, neither above-described applications of remote monitoring technologies can be realized without proper data quality. Therefore, only participants with a professional affinity for nutrition research and care were included. Indeed, the data from food logs appeared very complete, although this was not directly verifiable with reference data. Still, from the 24 participants, only 11 individuals had good-quality multimodal data for three consecutive days from the HowAmI app, the wristband, and the continuous glucose monitor (Figure 2). For the seven excluded participants, this was explained by specific problems regarding the ease of use of the research-grade wristband and the accompanying software, causing episodes of the device not collecting activity and sleep data in parallel to collecting glucose data. In particular, the software was primarily intended for researchers not for study participants and, therefore, not very user-friendly. Training and a 24/7 helpdesk were provided to anticipate the issues, but, unfortunately, this was not sufficient to obtain 100% data quality. Ideally, for a real-world design, data transfer is wireless without the need for active contributions from the participants. While there are devices available allowing such passive data collection and transfer, we still chose to use this device given its ability to collect raw data. Six participants were excluded because of incomplete continuous glucose monitoring data. While participants were blinded to the glucose data to make sure it did not influence their behavior, confounding the study results, they were also not able to observe whether actual data were collected. Hence, it was only after the study that these missing data were identified. On-device alarms on erroneous data collection could help participants to act earlier by replacing devices and improving continuous glucose data collection. Overall, these insights confirm the fact that results and outcomes from remote clinical trials strongly depend on data quality, correct use, and the connectivity of sensor technologies [24]. This stresses the need for easy-to-use digital devices in remote clinical trials [64,65]. Further remote investigations should expand on the current study, increasing sample size with a particular focus on easy-to-use digital devices. Another strength of the current research was the use of techniques to maximize personalized insights into contextual glucose relationships. A practical problem with machine learning models being used to capture the complex, non-linear relationships is their interpretation. The Shapley additive explanation (SHAP) approach was applied to explain the feature influence on the highest glucose levels for an individual. This approach is extensively utilized for explaining ‘black box’ machine learning models, allowing the calculation of the model features’ contribution to each individual data point [41]. Here, we selected the top 10 highest glucose levels to calculate the most important features contributing to those peaks for each individual. While, overall, activity-related features have a large influence on glucose levels, at an individual level, sleep or nutrition may be more important. Shapley values could, thus, form the basis for actionable insight into personalized lifestyle recommendations.

## 5. Conclusions

In this study, we explored the feasibility of data generated from current, wearable technologies for detecting eating moments and predicting the impact of physical activity, sleep, and dietary intake on continuous glucose levels in healthy volunteers. We showed that, pending further validation in a larger population, both eating moments and the influence of contextual lifestyle factors on glucose can potentially be predicted on an individual level. By opening up the ‘black box’ using SHAP, to our knowledge, this is the first study taking the step towards personalized, real-time lifestyle recommendations based on continuous health monitoring data. Eventually, the application of digital biomarkers that predict glucose from contextual factors is to drive personalized, continuous feedback on lifestyle factors to improve or maintain glucose homeostasis, thereby preventing the development of T2D.

The ease of use of wearable technologies is key for good data quality to allow for application in remote clinical trials, self-management, or remote care. Under everyday life conditions, we showed the feasibility of detecting eating moments to support food intake monitoring. Additionally, we showed how machine learning methods can be used to understand and explain individual relations between contextual lifestyle factors and interstitial glucose concentrations. Pending further validation, it is envisioned that these technologies will support self-management to maintain a healthy glucose metabolism through personalized lifestyle recommendations. Especially when combined with the early detection of insulin resistance and understanding of the biological cause for glucose derailment, the possibility exists that, in the future, meaningful digital biomarkers may provide the feedback and motivation to enable individuals to achieve the required lifestyle behavior change, ultimately allowing them to maintain health, prevent disease development, and reduce the economic burden of chronic diseases such as T2D.

## Figures and Tables

**Figure 1 nutrients-14-04465-f001:**
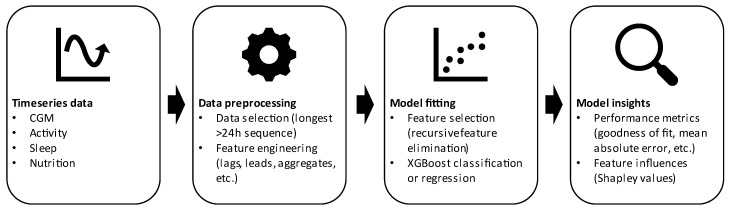
Schematic overview of the steps taken in model development and evaluation of the time-series data. Two models were developed, both following this workflow in a specific manner. The model predicting eating moments takes the CGM and activity time-series data as input to use XGBoost classification to classify whether there is an eating moment or not. The other model uses activity, sleep, and nutrition data as input to predict glucose levels, while, for individual glucose peaks, Shapley values are calculated to indicate the individual importance of activity, sleep, and/or nutrition in explaining these peaks. Further details are provided in the methods sections ‘Data preprocessing and feature engineering’ and ‘XGboost for predicting eating moments and glucose’.

**Figure 2 nutrients-14-04465-f002:**
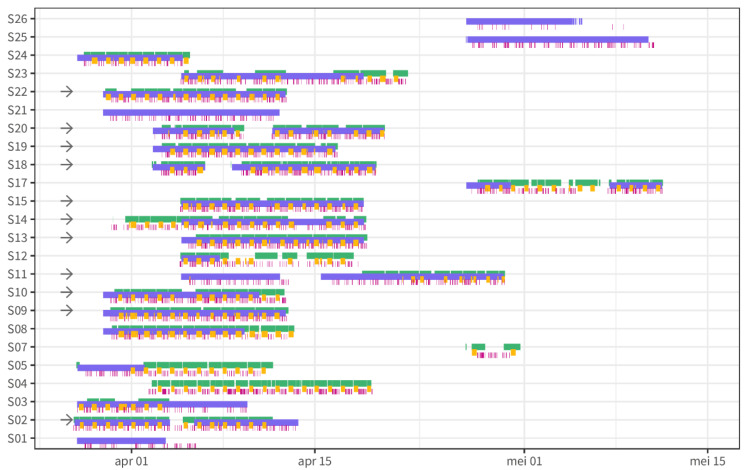
Overview of data availability for physical activity (green), sleep (yellow), dietary intake (pink), and interstitial glucose (purple). Data selected for further analysis are marked with an arrow.

**Figure 3 nutrients-14-04465-f003:**
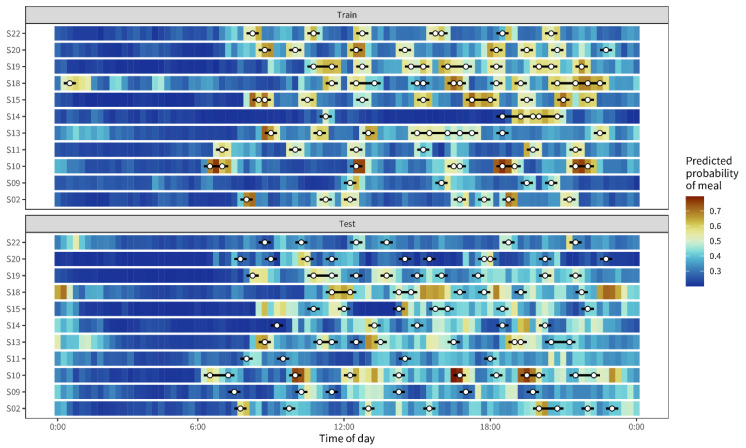
Predicted probability of an eating moment in blue against the indicated eating moments by the subjects (white dots). Probabilities were calculated for segments of 30 min; 15 min before and 15 min after an eating moment, indicated by the black bars around the white dots.

**Figure 4 nutrients-14-04465-f004:**
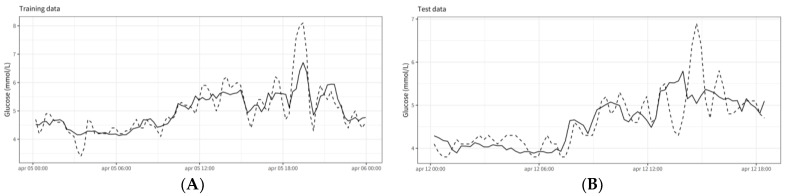
Goodness-of-fit model performance on example training data (**A**) and test data (**B**). Data come from subject 09.

**Figure 5 nutrients-14-04465-f005:**
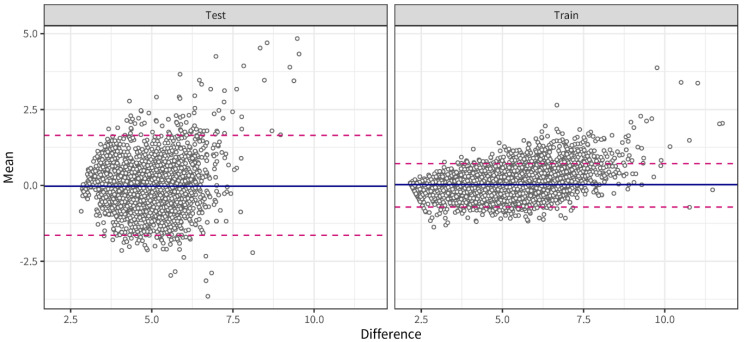
Bland–Altman analysis indicated a bias lower than 0.01 mmol/L in both the training and the test set and 2.5 and 97.5 percentile limits of agreement ranging from −0.72 to 1.1 in the training set and from −1.56 to 1.8 mmol/L in the test set.

**Figure 6 nutrients-14-04465-f006:**
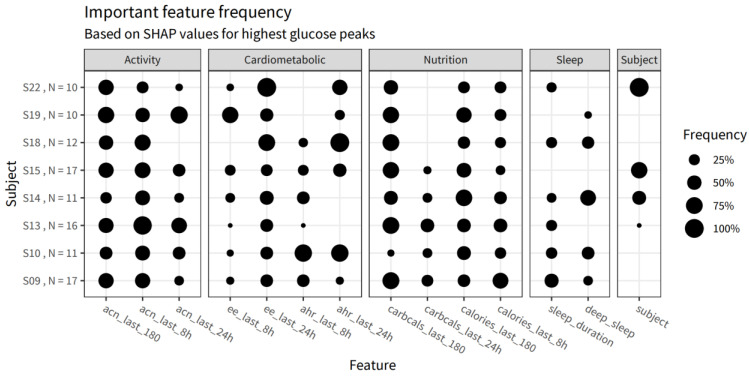
Frequency of use of the five most important features in the prediction of selected glucose peaks for individuals where more than 10 such peaks could be identified. The y-axis denotes the subjects and the number of peaks identified in each subject. ee: energy expenditure, acn: acceleration, ahr: average heart rate.

**Table 1 nutrients-14-04465-t001:** Overall feature influence of the different contextual modalities, activity (accelerometry), nutrition (carbohydrates, calories), and sleep (sleep duration, deep sleep duration), as well as cardiometabolic factors (energy expenditure, average heart rate) and a subject-specific factor.

Group	Weight
Cardiometabolic factors	26.7%
Subject	24.1%
Activity—Long term	12.5%
Nutrition—Short term	10.9%
Sleep	10.7%
Nutrition—Long term	8.7%
Activity—Short term	6.5%

## Data Availability

The datasets presented in this article are available upon reasonable request. Requests to access the datasets should be directed to corresponding author.

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
