# Peer review of "Digital Biomarkers for Personalized Nutrition: Predicting Meal Moments and Interstitial Glucose with Non-Invasive, Wearable Technologies"

_nutrients, 2022, doi:10.3390/nu14214465_

Round 1
Reviewer 1 Report
This is an interesting article. The area covered in this article is of my great interest as well.
Having said that, I find it bit difficult to follow what the authors have done and proposed, beyond the abstract. Even the abstract itself, it's unclear what is proposed and achieved.
The paper lacks any chart, table, algorithm, equation...nothing. Just a few graphs. The paper is very wordy for a technical article. It would have helped a lot if the authors had put together a flow chart and showed their proposed approach and solution, visually and with the help of mathematical expression.
Author Response
This is an interesting article. The area covered in this article is of my great interest as well.
Having said that, I find it bit difficult to follow what the authors have done and proposed, beyond the abstract. Even the abstract itself, it's unclear what is proposed and achieved.
The paper lacks any chart, table, algorithm, equation...nothing. Just a few graphs. The paper is very wordy for a technical article. It would have helped a lot if the authors had put together a flow chart and showed their proposed approach and solution, visually and with the help of mathematical expression.
We thank the reviewer for the constructive feedback that we have incorporated with the greatest possible care. First of all, the reviewer comments on the wordiness of the manuscript. Although this paper is not intended to be technical – rather we apply methods that have been developed and validated elsewhere – we agree that there is a lot of information in the text that may need some visual explanation in a flow chart. Indeed, in detail, we explained how features are engineered, models are fitted, and insight is created. Therefore, we have added a flowchart (figure 1) in addition to the (supplementary) tables and figures to provide a visual overview of the steps taken to apply these methodologies to gain insight into the relationship between contextual behavioral data and glucose dynamics. Specifically, the mathematical expression of XGBoost and SHAP is very lengthy and technical, and, in our opinion, does not contribute to communicating the work. Instead, we have shortly explained the idea of these methodologies and referred to seminal papers explaining them in more detail (Chen et al., 2021; Lundberg et al., 2020 - citations below). Second, it is unclear to the reviewer what is proposed and achieved. This work aims to develop digital biomarkers that provide insight into contextual lifestyle behavior, ultimately supporting health self-management and prevention of type 2 diabetes. We have further specified this in both the abstract and the introduction. Third, the reviewer points out that several aspects can/must be improved. Although it is not further specified what exactly is referred to we have self-evaluated the manuscript on these aspects and adapted it accordingly, with an applied research audience in mind.
Chen, T., He, T., Benesty, M., Khotilovich, V., Tang, Y., Cho, H., Chen, K., Mitchell, R., Cano, I., Zhou, T., Li, M., Xie, J., Lin, M., Geng, Y., & Li, Y. (2021). xgboost: Extreme Gradient Boosting. R Package Version 0.4-2 1.4, 1–4.
Lundberg, S. M., Erion, G., Chen, H., DeGrave, A., Prutkin, J. M., Nair, B., Katz, R., Himmelfarb, J., Bansal, N., & Lee, S.-I. (2020). From local explanations to global understanding with explainable AI for trees. Nature Machine Intelligence 2020 2:1, 2(1), 56–67. https://doi.org/10.1038/s42256-019-0138-9
Reviewer 2 Report
This article presents a multimodal sensory system associated with food logs, sleep, and activity tracking. It uses signal processing and pattern recognition algorithms to detect specific eating moments and predict blood glucose peaks using noninvasive monitoring with various commercial devices.
This topic is of great interest to health and nutrition management because type 2 diabetes causes many life-related problems. This system can provide a basis for personalized lifestyle advice.
The following recommendations are issued to the authors:
1. Figure 2 shows the prediction of an eating moment. Some moments (e.g., for S20 and S22, especially for S18) do not match between food diary and prediction. Is it possible to list why some predictions were not even close to a meal?
2. Figure 3 shows the performance of the glucose peak prediction based on the training and test datasets for subject 09 (although the prediction, according to Figure 2 for S09, was not optimal). There is some prediction discrepancy between the predicted and actual values. Although the MAE has a better evaluation value and Bland-Altman analysis has been performed, the metric evaluation of Mean Square Error is still needed since the outliers are actual or essential data and are the peaks that should be detected. Of course, if possible, the metric R^2 for the degree of model fit should also be calculated.
3. Because treeSHAP uses conditional prediction expectations and changes the value function, it can produce counterintuitive prediction results, i.e., features that do not affect the prediction result can receive a nonzero Shapley; and its effects are easy to misinterpret: It is the contribution value of the quality compared to its mean, not the yes/no difference. Therefore, the initial feature selection algorithm is more important, and only one feature selection method is used in this paper, RFE. If possible, please provide another feature selection method for comparison to check the effectiveness of the selected features.
4. The minimum length for continuous data was set to 24 hours. This is reasonable for appropriate data quality. Only 11 of the 24 subjects have suitable data quality. Only a brief explanation of low data quality is given. This should be further explained and elaborated. Over 50% of the subjects could not be evaluated due to a lack of data. Thus, this seems to be a significant problem of this study, which needs to be discussed in detail.
5. Blood glucose data from multiple meals may overlap if insufficient time has passed between the two meals. Likewise, exercise and sleep can affect blood glucose data. Was care taken to ensure at least 2 hours between different meals?
Author Response
This article presents a multimodal sensory system associated with food logs, sleep, and activity tracking. It uses signal processing and pattern recognition algorithms to detect specific eating moments and predict blood glucose peaks using noninvasive monitoring with various commercial devices.
This topic is of great interest to health and nutrition management because type 2 diabetes causes many life-related problems. This system can provide a basis for personalized lifestyle advice.
We would like to thank the reviewer for the constructive feedback that, we believe, truly improved the quality of the manuscript. Below, we respond point-by-point on how we have processed the feedback.
The following recommendations are issued to the authors:
- Figure 2 shows the prediction of an eating moment. Some moments (e.g., for S20 and S22, especially for S18) do not match between food diary and prediction. Is it possible to list why some predictions were not even close to a meal?
Unfortunately, we did not have a ground truth available for the eating moments because this study was performed in free-living conditions for which no ground truth measurement methods are available. Therefore, we used self-reported food logs as reference, which is subject to data collection errors. Discrepancies between observed and predicted eating moments may be caused by data collection errors (e.g., no reporting, erroneous time stamp) or by model inaccuracy. Overall, the data quality was high, in the sense that a lot of eating moments were reported, compared to our earlier experiences. Yet, we also observed lower specificity and sensitivity in the test set, indicating trouble generalizability of the model (see discussion section ‘Meal detection’). This problem could be approached by a more controlled approach including a ground truth referecence such as video camera monitoring of eating moments. Otherwise, a reinforcement learning approach may be applied in real-world settings to continuously update the algorithm within an individual, depending on the usecase. We have added these future directions now in the discussion section ‘Meal detection’.
- Figure 3 shows the performance of the glucose peak prediction based on the training and test datasets for subject 09 (although the prediction, according to Figure 2 for S09, was not optimal). There is some prediction discrepancy between the predicted and actual values. Although the MAE has a better evaluation value and Bland-Altman analysis has been performed, the metric evaluation of Mean Square Error is still needed since the outliers are actual or essential data and are the peaks that should be detected. Of course, if possible, the metric R^2 for the degree of model fit should also be calculated.
We fully agree that it is a good modeling practice to utilize model performance by multiple metrics. In fact, this is the reasons that we utilized both numeric and visual evaluations. Yet, we did want to be parsimonious in providing the reader with multiple metrics. Therefore, for publication, we carefully selected metrics that have most clinical relevance. When creating the regression model, we used the mean absolute error as a performance metric when evaluating model parameters in the cross-validation procedure. The MAE was chosen for this purpose because we judged this error to be the most meaningful from a practical perspective; the size of the absolute error in glucose values has real consequences for interpretation of predicted glucose value. For a similar reason, we applied Blant-Altman visualization for model evaluation. We did, however, calculate the MSE in addition to the MAE to evaluate model performance. (R² is mathematically equivalent to the MSE but while the MSE is proportional to the scale of the predictor, R² is not. In other words, R² is a scaled version of the MSE and, therefore, we believe this metric is not of additional value in our situation). We found that MAE and MSE differ little overall and therefore we decided to report only the MAE.
- Because treeSHAP uses conditional prediction expectations and changes the value function, it can produce counterintuitive prediction results, i.e., features that do not affect the prediction result can receive a nonzero Shapley; and its effects are easy to misinterpret: It is the contribution value of the quality compared to its mean, not the yes/no difference. Therefore, the initial feature selection algorithm is more important, and only one feature selection method is used in this paper, RFE. If possible, please provide another feature selection method for comparison to check the effectiveness of the selected features.
We fully agree variable selection is an essential step in the process and should therefore be considered carefully. In this specific work we did not compare different feature selection methods, since the main goal was to explore feasibility of the approach at all. We chose a single feature selection method so that we could demonstrate that the workflow shown here is one that will lead to a useable model. When further applying and scaling this approach, comparing multiple feature selection methods should be part of the modeling procedure. We do realize that this was not clearly communicated this limitation in the manuscript and have extended the discussion on this topic.
- The minimum length for continuous data was set to 24 hours. This is reasonable for appropriate data quality. Only 11 of the 24 subjects have suitable data quality. Only a brief explanation of low data quality is given. This should be further explained and elaborated. Over 50% of the subjects could not be evaluated due to a lack of data. Thus, this seems to be a significant problem of this study, which needs to be discussed in detail.
We appreciate the feedback from the reviewer asking for a more detailed explanation of low data quality and have extended the discussion on this topic. Also, we have better explained our criteria for excluding participants from analysis. While the quality of the food logging was surprisingly high in our experience, probably due to the fact that participants had affinity with nutrition research and care, there were issues with the watch and the CGM device. Participants had to offload the watch data in an accompanying software that was intended for researchers. Since this was a free-living study design and offload was needed on a daily basis, we asked participants to do the offload themselves after a short training and with 24/7 helpdesk. Even with these anticipatory actions, data from seven participants had to be excluded due to limited watch data. As for the CGM device, participants were blinded to the data to prevent confounding effects of changing behavior. Unfortunately, missing data was only observed after the study. These points have now been discussed in more detail in the discussion.
5. Blood glucose data from multiple meals may overlap if insufficient time has passed between the two meals. Likewise, exercise and sleep can affect blood glucose data. Was care taken to ensure at least 2 hours between different meals?
This study explicitly intended real-world conditions. Standardization was limited to a minimum, to as close to everyday life as possible. Therefore, no care was taken to ensure at least 2 hours between different meals. Moreover, while 2 hours difference may prevent short-term interactions, there are multiple longer-term interactions, such as type of food and exercise last 24 hours, sleep quality, etc. Therefore, we have engineered the features covering different rolling periods preceding glucose data, i.e., 30, 60, and 90 minutes as well as 2, 3, 8, and 24 hours, allowing modeling of the interactions between overlapping predictors of glucose data.
Round 2
Reviewer 2 Report
The manuscript has improved substantially. Publish it as it is.